# Additive Manufacturing Potentials of High Performance Ferritic (HiperFer) Steels

**Torsten Fischer [1],\*** 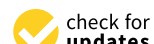**, Bernd Kuhn [1]** , **Xiuru Fan [1,2] and Markus Benjamin Wilms [3,4]**

1    Institute of Energy and Climate Research (IEK), Microstructure and Properties of Materials (IEK-2), Forschungszentrum Jülich GmbH, 52425 Julich, Germany; b.kuhn@fz-juelich.de (B.K.); x.fan@fz-juelich.de (X.F.)
2    Central Iron & Steel Research Institute (CISRI) Group, Material Digital R&D Centre, Beijing 100081, China
3    Materials Science and Additive Manufacturing, School of Mechanical Engineering and Safety Engineering, University of Wuppertal, 42119 Wuppertal, Germany; mwilms@uni-wuppertal.de
4    Fraunhofer-Institute for Laser Technology (ILT), 52074 Aachen, Germany
\*    Correspondence: t.fischer@fz-juelich.de; Tel.: +49-2461-61-5096

**Featured Application: Empowering metal additive manufacturing by unveiling the economic potentials of novel, low-cost alloys with improved mechanical properties (e.g., in comparison to conventional AM alloys, such as AISI 316L) and capable of in-build heat-treatment. Improvement of the quality of AM-processed material via the implementation of microstructural mechanisms for actively counterbalancing AM-process-related manufacturing flaws (pores, cavities).**

**Abstract:** In the present study, the first tailored steel based on HiperFer (high-performance ferrite) was developed specifically for the additive manufacturing process. This steel demonstrates its full performance potential when produced via additive manufacturing, e.g., through a high cooling rate, an in-build heat treatment, a tailored microstructure and counteracts potential process-induced defects (e.g. pores and cavities) via "active" crack-inhibiting mechanisms, such as thermomechanically induced precipitation of intermetallic $(Fe,Cr,Si)_2(W,Nb)$ Laves phase particles. Two governing mechanisms can be used to accomplish this: (I) "in-build heat treatment" by utilizing the "temper bead effect" during additive manufacturing and (II) "dynamic strengthening" under cyclic, plastic deformation at high temperature. To achieve this, the first HiperFer[AM] (additive manufacturing) model alloy with high precipitation kinetics was developed. Initial mechanical tests indicated great potential in terms of the tensile strength, elongation at rupture and minimum creep rate. During the thermomechanical loading, global sub-grain formation occurred in the HiperFer[AM], which refined the grain structure and allowed for higher plastic deformation, and consequently, increased the elongation at rupture. The additive manufacturing process also enabled the reduction of grain size to a region, which has not been accessible by conventional processing routes (casting, rolling, heat treatment) so far.

**Keywords:** additive manufacturing (AM); laser metal deposition (LMD); HiperFer[AM]; temperbead-effect; in-build heat treatment

---

## 1. Introduction

Up to now, mainly established commercial materials have been applied in additive manufacturing. Materials that have been tailored for AM processes (including mutual process optimization) are still underrepresented in international R&D efforts. The main reasons for this are high development expenses and a still relatively limited market. In order to improve the applicability of AM, the economic efficiency of both the materials and processes have to be increased. In this paper, a material is presented that benefits from the additive manufacturing process and can be customized to desired applications not only by varying chemical composition but also by variations in AM process parameters. Process parameters,



such as beam diameter, laser power and scan speed, determine the shape and size of the melt pool formed by alternating the laser absorption ratios [1,2]. Solidification conditions of the melt pool are controlled by the heat dissipation mechanisms [3], which are influenced by material properties, geometrical aspects and temperature distribution in the printed material [4]. The characteristic high solidification rates encountered in additive manufacturing processes typically result in the formation of supersaturated solid solutions, making a post-process heat treatment necessary [5]. However, in additive manufacturing, adjustments of the process parameters may induce in-process precipitation reactions [6–9], phase transformations [10,11], affect residual stress in the printed parts [12–15], and modify the grain structure and texture [5,16,17]. Thus, mechanical properties, e.g., tensile and fatigue strength, can be significantly altered by the utilization of different process parameters [14,18–21]. Process parameters also have a decisive impact on defect formation, the final density, and thus, the material integrity. Residual porosity is a key factor for the mechanical and technological properties of additively manufactured materials because process-related pores [21–24] can act as crack initiators [9]. So far, the approach to overcoming the residual porosity is the optimization of process control. Advanced detectors and monitoring software are being developed to control production quality [9]. Often post-treatment procedures, such as hot isostatic pressing, are necessary to reduce porosity, which constitute an economical drawback.

Partially counteracting process-related material defects by utilizing material's inherent microstructural crack-obstruction mechanisms, so that component defects can be tolerated to a certain extent, would be a highly innovative approach. The novel, high-chromium ferritic HiperFer (high-performance ferrite) steels developed at Forschungszentrum Jülich GmbH [25,26] are suitable material candidates to achieve this for several reasons: strengthening via a combination of solid solution and precipitation hardening by intermetallic Laves phase particles ($(Fe,Cr,Si)_2(Nb,W)$) [25]. "Dynamic strengthening" of HiperFer is accomplished via thermomechanically induced precipitation [27] of the Laves phase using conventional processing [28–31] under in-service conditions [32,33]. How effectively crack initiation is inhibited/delayed in Laves phase strengthened ferritic steel was demonstrated using thermomechanical fatigue (TMF) experiments on Crofer®22 H, the predecessor alloy of HiperFer (cf. [33] for details). The fatigue curve of Laves-phase-strengthened steel typically consists of three phases [33]: an initial phase of pronounced strengthening (which makes up for about 1% of the fatigue life), a stable phase (which lasts until approximately half of the fatigue life) and a comparatively protracted damage phase (making up for approximately the other half of fatigue life). During the damage phase, a comparatively small decrease in stress range per cycle is observed because cyclic plasticization dislocations are generated, leading to both non-permanent strain hardening and accelerated nucleation of intergranular Laves phase particles. As a result of the effective blocking of the dislocations by the deformation-induced Laves phase particle precipitation (under the experimental conditions lined out in [33], the total number of Laves phase particles in Crofer®22 H, for example, increased from 0 to 4576 during testing from 10 to 2000 TMF cycles), part of the dislocation-induced strain hardening remains practically permanent. For a detailed explanation of the mechanisms and the complete evolution of intragranular Laves phase precipitates in Laves-phase-strengthened ferritic steel during TMF loading, [33,34] may be consulted.

Furthermore, high chromium ferritic steels generally possess good resistance to (high-temperature) corrosion due to the formation of a protective duplex $Cr_2O_3/(Mn, Cr)_3O_4$ scale on the component surface [35]. The chromium content of HiperFer was adjusted to 17 wt.% to exclude the undesired formation of the FeCr σ-phase at temperatures above 600 °C [25], and thus, the potential danger of embrittlement due to long-term service at elevated temperatures.

With this property profile, an alloy that is custom designed for additive manufacturing on the basis of HiperFer is a good candidate to replace typical austenitic grades, such as AISI 316L, at much lower material and processing costs. A suitable way to make the dynamic strengthening properties of HiperFer steel accessible for in-build and in-service

strengthening of AM-processed components is tuned precipitation kinetics. Based on the outlined mechanisms, a first HiperFer[AM (additive manufacturing)] alloy was created specifically for additive manufacturing processes, which ideally should utilize both the "dynamic strengthening" and the so-called "temper bead-effect" [36–39] for in-build heat treatment (to make downstream heat treatments obsolete). In this instance, a suitable in-build heat treatment was aimed at keeping the residual stresses below the yield strength of the material and simultaneously generating Laves-phase-strengthened precipitates. In addition, the paper outlines the obtained mechanical property data, such as tensile strength, elongation at rupture and minimum creep rates (depending on the build direction). Furthermore, the active deformation mechanisms in additively processed material were compared to conventionally manufactured (casting, rolling, heat treatment) material.

## 2. Materials and Methods

### 2.1. Laves-Phase-Strengthened Ferritic HiperFer Steel

The HiperFer trial steel 17Cr2 was vacuum induction melted (by the Steel Institute of RWTH Aachen University, Germany). The resulting ingot was hot-rolled to 15 mm thick plate material in the temperature range from 1000 to 920 °C and subsequently precipitation annealed (0.5–10 h/600–650 °C). The chemical compositions of HiperFer 17Cr2 and laser metal deposition (LMD)-manufactured HiperFer[AM] are given in Table 1.

**Table 1.** Chemical compositions of Laves-phase-strengthened ferritic trial steels (wt.-%).

| | C | S | N | Cr | Ni | Mn | Si | Mo | Ti |
|---|---|---|---|---|---|---|---|---|---|
| **22 A** | 0.004 | <0.001 | 0.002 | 22.8 | - | 0.44 | 0.02 | - | 0.06 |
| | Nb | W | Cu | Fe | P | Al | Mg | Co | La |
| | <0.01 | - | - | **Balance** | - | 0.013 | - | - | - |
| | C | S | N | Cr | Ni | Mn | Si | Mo | Ti |
| **HiperFer 17Cr2 *** | <0.003 | <0.001 | <0.003 | 17–18 | - | 0.2–0.5 | 0.2–0.3 | - | - |
| | Nb | W | Cu | Fe | P | Al | Mg | Co | La |
| | 0.5-0.6 | 2.4-2.6 | - | Balance | - | - | - | - | - |
| | C | S | N | Cr | Ni | Mn | Si | Mo | Ti |
| **HiperFer[AM]** | 0.0033 | <0.0008 | 0.0017 | 19.152 | - | 0.365 | 1 | - | 0.05 |
| | Nb | W | Cu | Fe | P | Al | Mg | Co | La |
| | 1.5 | 2 | - | Balance | - | - | - | - | - |

* [40].

### 2.2. LMD-Manufactured HiperFer[AM]

The base HiperFer[AM] powder material was manufactured from gas-atomized experimental 22 A powder (chemical composition listed in Table 1). Since 22 A powder was utilized as the base composition, the first HiperFer[AM] alloy featured a higher chromium content than HiperFer 17Cr2 (cf. Table 1).

The final powder material was manufactured by blending the 22 A powder with high-purity Si, Nb and W powders in a Turbulaσ-mixing device (WAB GmbH) for 30 min. As a substrate material, a tool steel (AISI H10, 1.2365) was used, which was sand-blasted and cleaned with ethanol prior to deposition. LMD processing was conducted with two individual 5-axis handling systems, both using fiber-coupled diode lasers (Laserline GmbH, Mülheim-Kärlich, Germany). For the large beam diameter (3 mm), also referred to as the low "power density" in Table 2, an LDM3000-60 laser, emitting laser radiation with a wavelength of 976 nm (beam parameter product (BPP): 60 mm*mrad), was used. The laser beam was coupled into an optical fiber with a core diameter of 600 μm (NA = 0.2) and subsequently shaped using collimation ($f_{c, 3 mm}$ = 65 mm) and focusing lenses ($f_{f, 3 mm}$ = 230 mm). The smaller beam diameter (0.66 mm) was created by using an LDF2000-30 diode laser system, simultaneously emitting laser radiation with 1025 nm and 1064 nm. Beam shaping after the optical fiber (core diameter: 600 μm, NA = 0.2) was enabled using a combination of collimation ($f_{c, 0.66 mm}$ = 200 mm) and focusing lenses ($f_{f, 0.66 mm}$ = 182 mm). The powder

material was fed pneumatically with argon gas via a disc-based feeding system (Sulzer-Metco Twin 10C, OC Oerlikon AG, Pfäffikon, Switzerland) through a coaxial 3-jet nozzle (D40-type, Fraunhofer ILT, Aachen, Germany). For melt pool shielding, argon gas was used, which was fed through the tip of the coaxial powder nozzle. The HiperFer$^{AM}$ powder blend was processed with the process parameters shown in Table 2.

**Table 2.** HiperFer$^{AM}$ LMD process parameters.

| Process Parameter | Low Power Density ("LPD") | High Power Density ("HPD") |
|---|---|---|
| Area-specific laser power | 195 Wmm$^{-2}$ | 1050 Wmm$^{-2}$ |
| Feed speed | 400 mm/min | 1500 mm/min |
| Laser power | 1375 W | 360 W |
| Powder mass flow rate | 11.3 g/min | 2.4 g/min |
| Track offset | 2.3 mm | 0.35 mm |
| Height offset | 1.4 mm | 0.3 mm |
| Beam diameter | 3 mm | 0.66 mm |
| Deposition strategy | Unidirectional | Unidirectional |
| Nozzle type | 3-jet nozzle | 3-jet nozzle |
| Shielding gas | Argon | Argon |

### 2.3. Mechanical Testing

Tensile experiments at ambient temperature were conducted on miniature size specimens (thread size: M5, l = 42 mm, $l_0$ = 20 mm and $d_0$ = 3 mm) according to DIN EN ISO 6892-1 (constant strain rate $\dot{\varepsilon} = 10^{-3}\,S^{-1}$) [41] utilizing an Instron 1381 type testing machine with 10 kN of load capability. At 650 °C, tensile tests were carried out according to DIN EN ISO 6892-2 [42]. In accordance with the standard, a strain rate of $\dot{\varepsilon}_{el} = 8.33 \times 10^{-5}\,S^{-1}$ was applied in the elastic deformation range, while in the plastic range, a strain rate of $\dot{\varepsilon}_{el} = 8.33 \times 10^{-4}\,S^{-1}$ was utilized. All tensile specimens were electrically discharge machined (EDM) from the HiperFer$^{AM}$ LMD samples in the building (Z) direction (Figure 1).

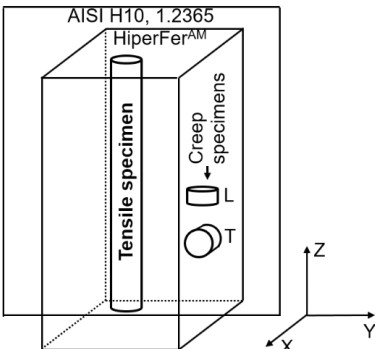

**Figure 1.** Schematic representation of electrical discharge machining of mechanical testing specimens from HiperFer$^{AM}$ LMD samples: creep specimens were taken from both the longitudinal and transversal directions, while tensile specimens were taken from the longitudinal direction only.

Stepped increasing stress (70, 100, 120, 130 MPa) compression creep experiments were performed on miniature cylindrical specimens (d: 3 mm, h: 3.5 mm) at 650 °C by applying an Instron 1381 testing machine (Instron, Darmstadt, Germany) that was equipped with all-ceramic loading and strain measurement set-ups. The testing stress was incrementally increased upon entry of a quasi-secondary creep stage or after 150 h of holding at each stress level. Temperatures were controlled to an accuracy of +/− 1 °C using type S thermocouples attached to the specimens' gauge length in the creep experiments. The strain measurement was performed with a digital 5 mm strain transducer (Solartron, accuracy: 1.1 μm (peak to peak error)). The compression creep specimens were EDM from the longitudinal (L)

and transversal (T) directions in relation to the build direction from the HiperFer[AM] LMD samples (Figure 1).

Vickers hardness tests (HV1) were performed according to DIN EN ISO 6507 [43] by applying a Qness A10+ hardness tester (ATM Qness GmbH, Mammelzen, Germany).

### 2.4. Microstructural Investigation

Samples for microstructural analysis were cut from the HiperFer[AM], hot embedded, ground and polished to a sub-micron finish in a colloidal silica suspension and $Al_2O_3$ in dilute KOH solution for approx. 4 h. Electrolytic etching at 1.5 V in 5% $H_2SO_4$ was subsequently performed to enhance the particle/matrix contrast. A Zeiss Merlin field emission scanning electron microscope (FESEM) was utilized for high-resolution microstructural investigations.

The small size of the early stage Laves phase precipitates and the magnetic properties of HiperFer steels make detailed analysis using traditional methods inaccurate and time-consuming. For this reason, particle analysis was performed using quantitative image analysis. Particle analysis was conducted at 2 representative sample positions. Grain boundaries and particle-free zones were excluded in the determination of particle density. The size distributions of Laves phase particles were determined using "background corrected" SEM images (accomplished applying the "Count and measure for Olympus Stream" package in the Olympus Stream Desktop 2.4 commercial software). Subsequently, the mean value of the data was used as the analysis result. A detailed description of the particle analysis is given in [44].

Additionally, samples for EBSD analysis were prepared from the mechanical testing specimens. EBSD analysis was performed using a Zeiss Merlin SEM equipped with an Oxford Instruments EBSD system (NORD LYS 2 camera) and energy dispersive X-ray spectroscopy (EDS, EDX, X Max 150).

### 3. Results and Discussion

HiperFer steel owes its strength to a combination of solid solution strengthening and precipitation hardening using thermodynamically stable $(Fe, Cr, Si)_2(Nb, W)$ Laves phase particles [25]. Nb and Si are key elements in Laves phase precipitation: Nb is used because when combined in an alloy with tungsten, it improves solid solution strengthening to some extent [45]. Furthermore, it is a strong Laves phase former [46], and because of its comparatively low solubility and high diffusion rate, it facilitates precipitation in ferrite. Si further accelerates nucleation [46–50] of the Laves phase [25,35,46,51] precipitates. Figure 2 depicts the solid solution strengthening effects of W and Nb in HiperFer steel. The figure indicates a more pronounced impact of W on UTS and a greater impact of Nb on $YS_{0.2}$.

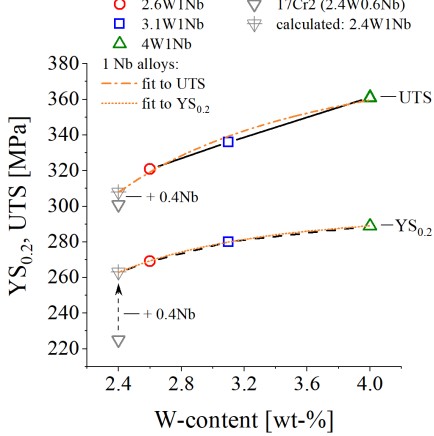

**Figure 2.** Impact of W and Nb in solid solution on ambient temperature tensile strength values with a virtual 2.4W1Nb alloy, equation: $YS0.2 = a \times (1 - cW - b)$; cW: W-content in wt.%, a: 300.05, b: 2.38; $UTS = a \times (1 - cW - b)$; cW: W-content in wt.%, a: 396.53, b: 1.71 [31].

In the case of a 17Cr2 base alloy (2.4W0.6Nb), $YS_{0.2}$/UTS values of 400/660 MPa were achieved in the precipitation heat-treated state [33].

Increasing the W and Nb contents improved the solid solution strengthening effect, along with increasing the volume fraction and reducing the average particle size of the Laves phase precipitates, resulting in enhanced mechanical properties, in particular creep strength [31]. Figure 3 displays the effects of varying the W and Nb contents on the calculated limit stress for dislocation creep $\sigma_{th}$ (for details cf. [31,52]): using W and Nb alloying, an increase in the limit stress of approximately 61% was achievable.

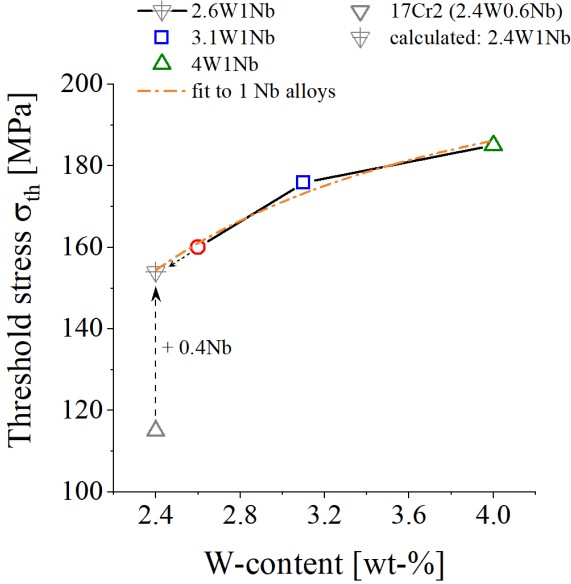

**Figure 3.** Impact of W and Nb on the threshold stress $\sigma_{th}$ [31].

The addition of Si mainly resulted in a reduction in the Laves particle size, increased the number of particles and accelerated the precipitation kinetics [53]. From the evaluated increase in limit stress for dislocation creep $\sigma_{th}$ by Si-alloying (Figure 4), it could be derived that by Si alloying, an approximate increase of 43% was obtainable.

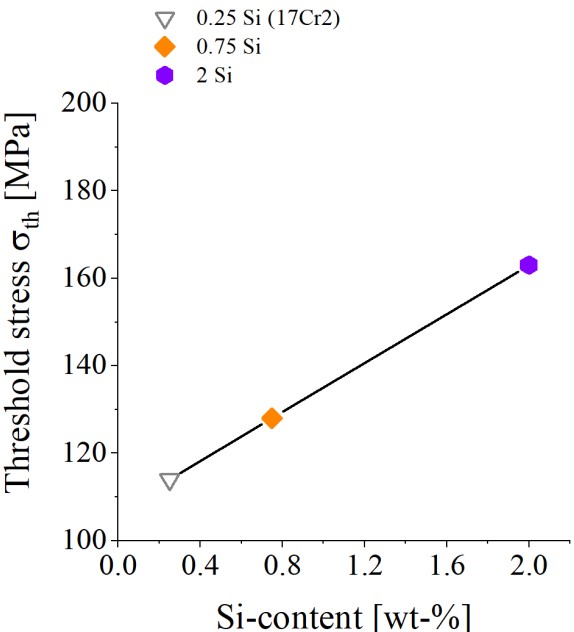

**Figure 4.** Effect of Si on $\sigma_{th}$ [54].

Based on these results, the Nb/Si contents were adjusted to 1.5/1 wt.% in order to maximize the (i) precipitation kinetics and (ii) volume fraction and to decrease the particle size in the HiperFer$^{AM}$ (Table 1) trial alloy. Because of the high melting point discrepancy between W and the other constituents, a moderate content of W (2 wt.%) was chosen to alleviate the expected difficulties in the additive manufacturing process.

The increased Nb and Si contents (compared with conventionally casted, rolled and heat-treated HiperFer 17Cr2 (Table 1)), in combination with the high cooling rates of the AM process, resulted in a highly supersaturated solid solution alloy matrix in the as-built state, which in turn led to rapid precipitation kinetics as a result of the high precipitation pressure toward equilibrium. Despite comparatively short high-temperature periods, the in-build (i.e., temper bead) heat treatment (Figure 5) successfully produced a large amount of finely dispersed intragranular Laves phase particles. In addition, the high angle grain boundaries (HAGb) were covered by Laves phase precipitates, which stabilized the grain structure without embrittling the material. The stabilization of the grain structure is the key factor that allows for dynamic strengthening [27,55] under fatigue loading. Dynamic hardening/precipitation was successfully exploited in aluminum alloys AA2021, AA6061 and AA7050 to enhance high-cycle fatigue performance [56]. For this purpose, additional fine particles were generated in the particle free zones (PFZs) using suitable cyclic training. In the fully precipitated state (highest strength state), no dynamic precipitation was observed under cyclic loading in these aluminum alloys. This, however, was achieved in the HiperFer and HiperFer$^{AM}$ type alloys.

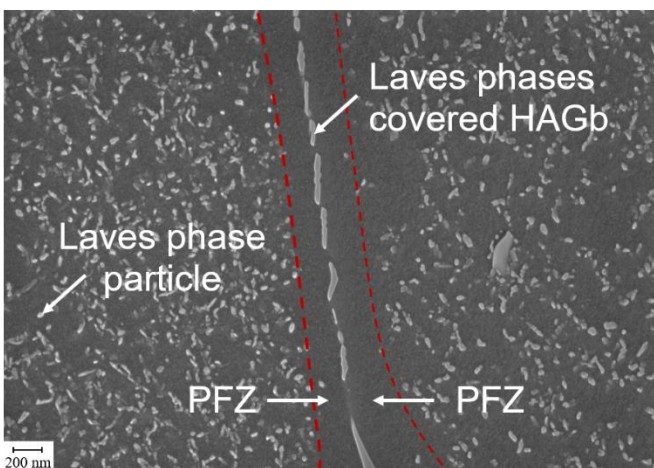

**Figure 5.** Microstructural features of LMD-manufactured HiperFer$^{AM}$ (LPD variant, further process parameters, cf. Table 2).

Process-induced pores that are contained in additively manufactured materials can act like notches and cause an accumulation of plastic strain in the surrounding material under cyclic loading. In steels of the HiperFer type, this induces thermomechanically induced precipitation [28,32] of Laves phase particles at these dislocations at high temperature, which strengthen the vicinity of pores and actively obstruct crack propagation.

Within the scope of this study, it was investigated to which degree microstructure characteristics (grain size, particle number, size, etc.) can be influenced "in-build" by varying the AM process parameters. For this purpose, the area-specific laser powers of 195 and 1050 W/mm$^2$ were chosen. HiperFer$^{AM}$ manufactured with a specific laser power input of 195 W/mm$^2$ in the following is referred to as the low-power-density (LPD) variant, while an energy input of 1050 W/mm$^2$ was the high-power-density (HPD) variant. Further AM parameter variations are summarized in Table 2.

The typical microstructure of recrystallized and quality heat-treated ferritic, Laves-phase-strengthened steels consists of globular grains with particle-free zones (varying from 200–2000 nm in width, depending on the chemistry and temperature/stress history) along

HAGbs (Figure 6). Near the PFZ ("IG$_{Exterior}$" region in Figure 6), the particles were usually slightly smaller. Gradual particle dissolution was argued to be the reason for this particular feature. The elements dissolving into the matrix diffuse toward the grain boundary and gradually coarsen the Laves phase particles located there. Accordingly, the particles in the grain interior ("IG$_{center}$" region in Figure 6) were found to be larger [11].

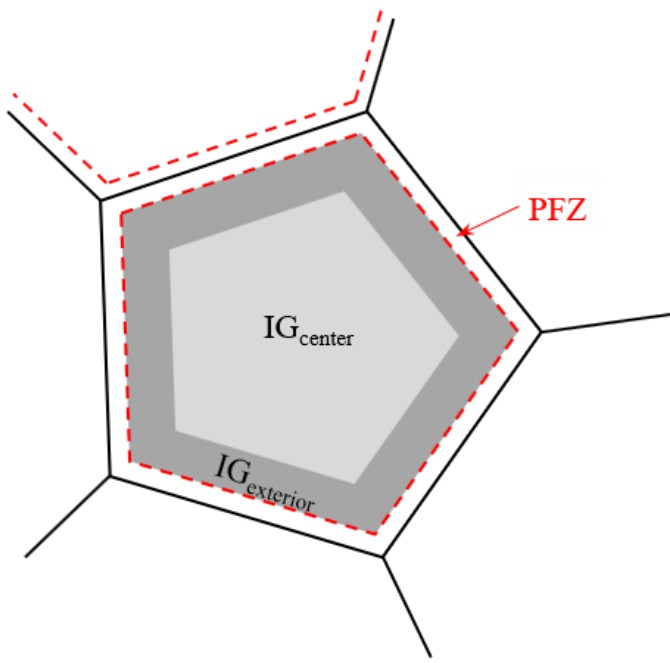

**Figure 6.** Schematic representation of the typical microstructure of recrystallized and quality heat-treated ferritic Laves-phase-strengthened steel [30].

Large, aligned, rod-shaped grains were predominantly observed in the LPD-manufactured HiperFer$^{AM}$, while only a small area of aligned, even smaller rod-shaped grains was encountered in the HPD variant (Figure 7) where the globular grain shape prevailed (Figure 7b). This demonstrated that the grain size could readily be adjusted by varying the AM processing parameters. Conventionally processed HiperFer 17Cr2 typically prevails in a grain size of approx. 500 μm, while both applied AM parameter sets resulted in predominantly globular grains in the size region of approx. 200 μm and less (Figure 8). However, incompletely melted tungsten clusters were observed (cf. Figure 7). An additional processing-related issue was SiO$_2$ particles. These particles tended to be larger and more numerous in the LPD variant than in the HPD variant (cf. Figure 7). Finley-distributed SiO$_2$ particles were also observed in LPBF-manufactured 316L [9].

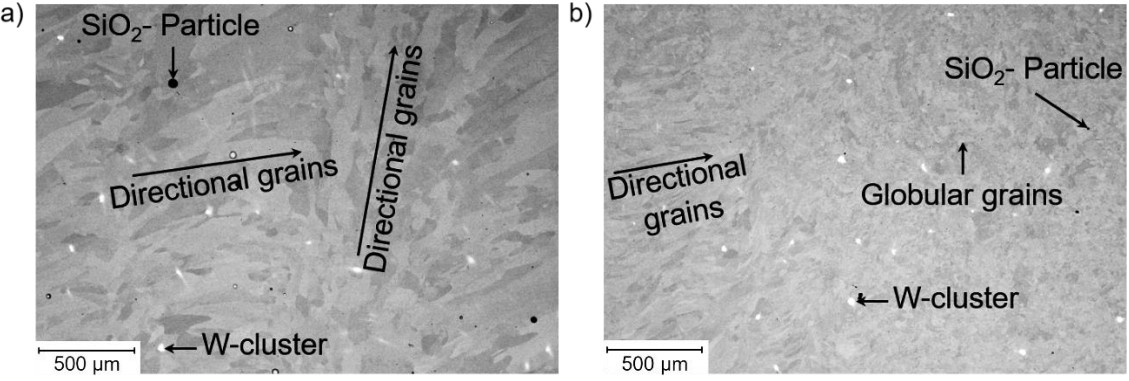

**Figure 7.** SEM micrographs of LMD-manufactured HiperFer$^{AM}$ in the as-built state: (**a**) LPD and (**b**) HPD variants.

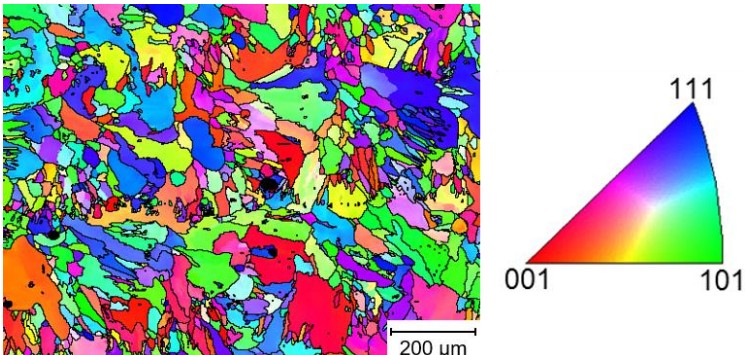

**Figure 8.** EBSD-IPF (inverse pole figure) micrograph of the LMD-manufactured HiperFer[AM] HPD variant in the as-built state (reproduced from [9]).

In the future, this can be prevented via the preparation of a master alloy from which the AM powder will be produced. Moreover, the change in power density resulted in a different number and size of Laves phase particles (cf. Figure 9). In the LPD material, the area-specific total number of particles in the as-built state was more than twice that of the HPD variant (cf. Figures 9 and 10a). The particle size classification demonstrated a clear tendency toward larger particle diameters in the HPD variant. The proportion of particles below 20 nm was lower by more than a factor of 5 in the HPD variant (Figure 10b). The lower power input explained the tendency toward the smaller particle size in the case of the LPD variant well. In addition to the higher power input, a smaller beam diameter was applied in the case of the HPD manufactured material. For this reason, more surface area was re-melted and material cooling was slower, i.e., more heat and time were available for particle growth.

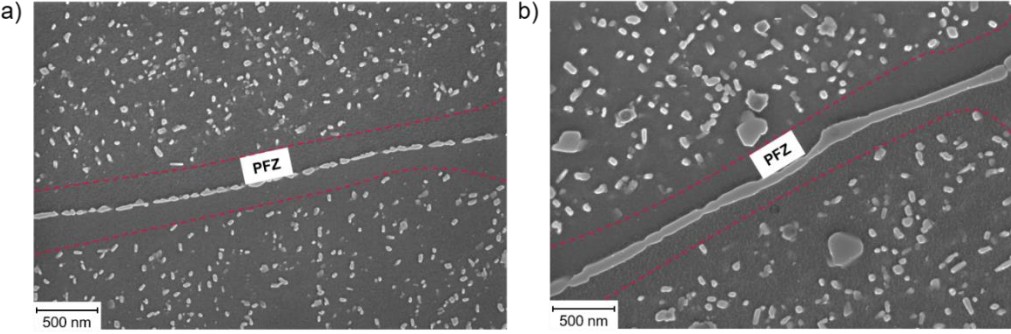

**Figure 9.** Laves phase particles in LMD-manufactured HiperFer[AM]: (**a**) LPD and (**b**) HPD variants.

In conventionally fabricated HiperFer 17Cr2 (recrystallized and precipitation annealed for 2 h at 650 °C), the highest area-specific particle number was observed (Figure 10a). Considering the time necessary for Laves phase precipitation, the advantage of HiperFer[AM] clearly emerged: While the heat treatment of HiperFer 17Cr2—including recrystallization and precipitation annealing—took more than 2 h, the heat treatment of HiperFer[AM] was carried out in situ despite re-melting in the range of seconds. In HiperFer 17Cr2, as in the LPD manufactured HiperFer[AM], particle fractions smaller than 20 μm and between 20 and 40 μm were formed (Figure 10b).

The width of the PFZs (approximately 300 nm) in HiperFer[AM] was found to be independent of the area-specific laser power. Larger parameter field studies are required to evaluate to what extent the PFZ width can be tuned by the in-build heat treatment, which may have a direct effect on material ductility.

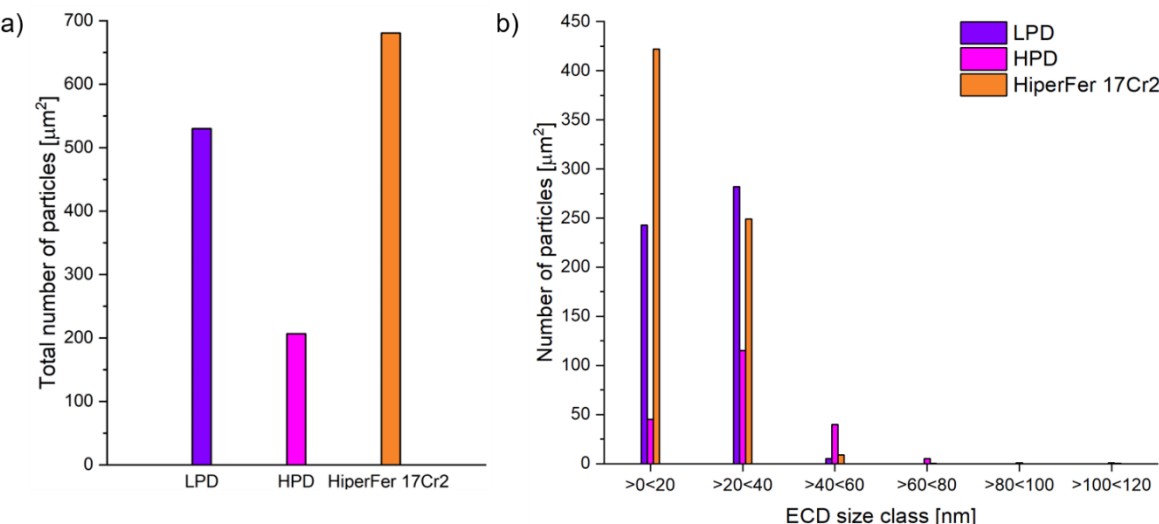

**Figure 10.** (**a**) Area-specific total number of particles in the as-built LPD and HPD HiperFer[AM] and conventional HiperFer 17Cr2 Pa (2 h 650 °C) materials and (**b**) corresponding size classification by equivalent circle diameter (ECD).

Regions of varying particle size, such as the "$IG_{Exteriour}$" and "$IG_{center}$" regions typically encountered in conventionally fabricated HiperFer (Figure 6), were not observed in HiperFer[AM]. This was caused by the shorter cumulated time at high temperatures and the absence of soaking.

Tensile tests were carried out to characterize the macroscopic impact of the microstructures generated in-build. The resulting UTS and total strain at rupture are depicted in Figure 11, while the corresponding values are listed in Table 3. In comparison to conventionally (rolled, recrystallized, precipitation annealed (650 °C/2 h)) produced 17Cr2 sheet material, the ambient temperature tensile strength values of LMD-processed HiperFer[AM] were found to be higher ($UTS_{HPD}$: + 107 MPa, $UTS_{LPD}$: + 14 MPa).

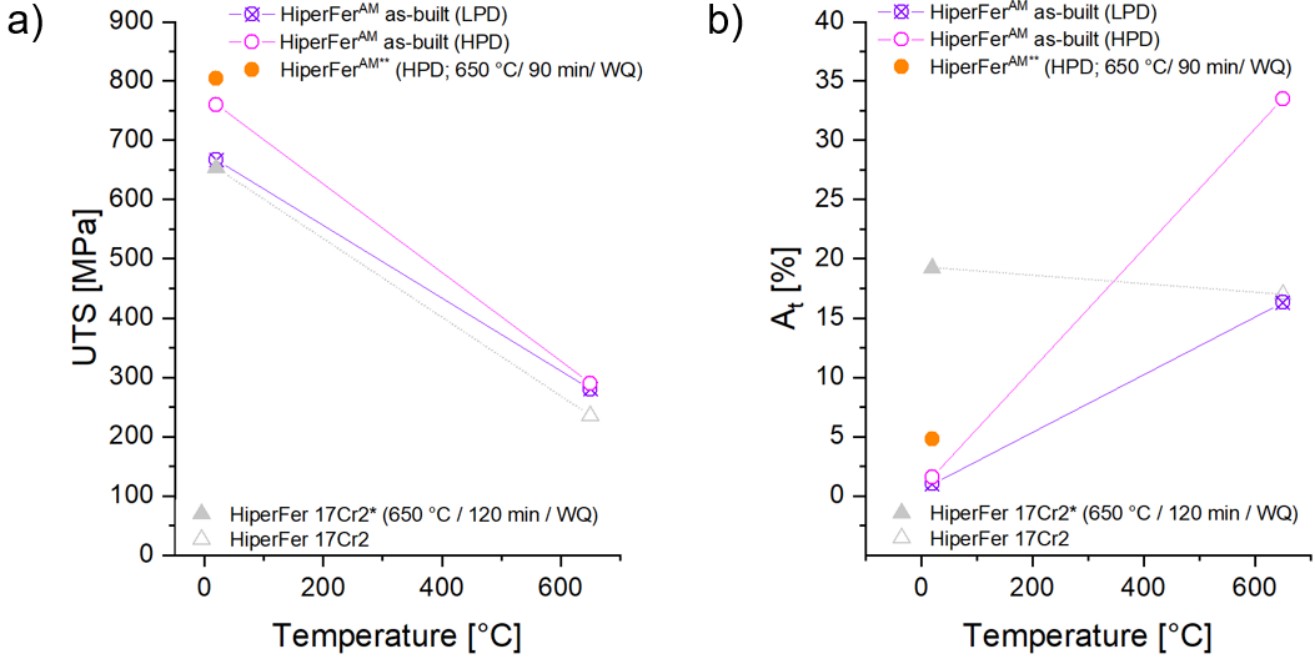

**Figure 11.** (**a**) UTS and (**b**) rupture elongation ($A_t$) of the 17Cr2 and HiperFer[AM] variants (LPD and HPD) at 20 °C and 650 °C. HT: * 650 °C/120 min, ** 650 °C/90 min.

**Table 3.** UTS (20 and 650 °C) values and A$_t$ of HiperFer 17Cr2 compared with the HiperFer$^{AM}$ LPD and HPD variants.

| Material | Heat-Treatment State | Area-Specific Laser Power (Wmm$^{-2}$) | Test Temperature (°C) | UTS (MPa) | A$_t$ (%) |
|---|---|---|---|---|---|
| HiperFer 17Cr2 | 650 °C/120 min | - | 20 | 653 | 19.2 |
| HiperFer 17Cr2 | - | - | 650 | 235 | 17 |
| HiperFer$^{AM}$ | As-built | 1050 (HPD) | 20 | 760 | 1.6 |
| HiperFer$^{AM}$ | As-built | 195 (LPD) | 20 | 667 | 1 |
| HiperFer$^{AM}$ | 650 °C/90 min | 1050 (HPD) | 20 | 804 | 4.8 |
| HiperFer$^{AM}$ | As-built | 1050 (HPD) | 650 | 290 | 33.5 |
| HiperFer$^{AM}$ | As-built | 195 (LPD) | 650 | 280 | 16.3 |

The advantage of HiperFer$^{AM}$ (HPD) over HiperFer$^{AM}$ (LPD) in terms of the UTS was caused by the smaller grain size (Figure 12) and the increased level of residual stress (i.e., higher area-specific power input). The ambient temperature rupture elongations of both the additively manufactured variants had expectedly low values of 1% (LPD) and 1.6% (HPD). In contrast, conventionally produced 17Cr2 yielded ~17% fracture elongation. In accordance with the encountered low fracture elongations at ambient temperature, the EBSD mappings of the additively manufactured tensile specimens present hardly any deformation (Figure 12), which was most likely caused by a comparatively high level of residual stress that resulted from the additive manufacturing process (hardness LPD/HPD 199/226 HV1, cf. Table 4).

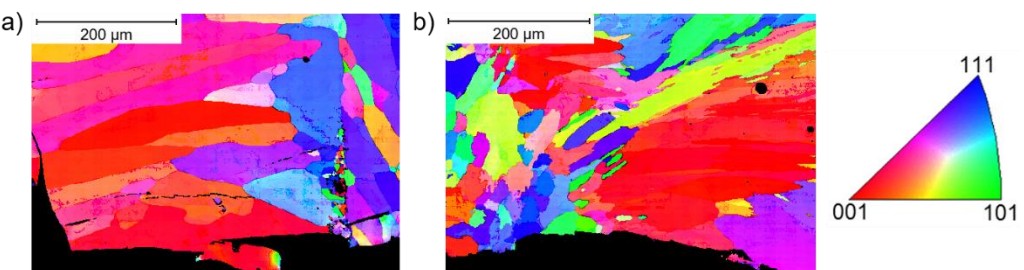

**Figure 12.** HiperFer$^{AM}$ (**a**) LPD and (**b**) HPD EBSD mappings taken perpendicularly to the fracture surfaces after the tensile experiments at ambient temperature.

**Table 4.** Hardnesses of LPD and HPD HiperFer$^{AM}$ in the as-built state and after compression creep testing.

| Material | Hardness, As-Built State (HV1) | | | | Hardness, After Creep Compression Test (HV1) | | | |
|---|---|---|---|---|---|---|---|---|
| | Mean | Min. | Max. | Range | Mean | Min. | Max. | Range |
| HiperFerAM LPD-L | 199 | 193 | 209 | 16 | 211 | 197 | 222 | 25 |
| HiperFerAM HPD-L | 226 | 217 | 233 | 16 | 248 | 239 | 252 | 13 |
| HiperFerAM LPD-Q | 196 | 186 | 201 | 15 | 218 | 206 | 226 | 20 |
| HiperFerAM HPD-Q | 233 | 217 | 244 | 27 | 252 | 240 | 272 | 32 |

To evaluate whether high residual stress was the root cause for the low ambient temperature fracture strain, additively manufactured (HPD) tensile specimens were heat-treated at 650 °C for 90 min and subsequently tensile tested at ambient temperature. This heat treatment increased the fracture elongation by an approximate factor of 3 to a value of ~5%, while the UTS further increased to about 800 MPa (cf. Figure 11 and Table 3).

A conclusive assessment of the high level of residual stress as the reason for the low ambient temperature ductility was not possible because any heat treatment for stress reduction potentially led to the precipitation of additional strengthening Laves phase particles.

However, in view of the tensile properties at 650 °C, this assumption was reasonable. At 650 °C, HiperFer[AM] (HPD) reached UTS/fracture elongation values of 290 MPa/33.5%, which constituted a remarkable improvement over conventionally produced HiperFer 17Cr2 sheet material (UTS: 235 MPa, $A_t$: 17%, cf. Figure 11 and Table 3). Concerning the UTS, no significant difference existed when applying the LPD parameters (cf. Figure 11 and Table 3), but the total fracture elongation diminished to about 16%, which still correlated well with the conventionally rolled sheet material. In the LPD HiperFer[AM], localized sub-grain formation was observed (Figure 13a), while global sub-graining occurred in the HPD variant (Figure 13c). Since the deformation conditions in the tensile test were identical, a higher stacking fault energy (i.e., the ability of dislocations for transversal slip) in the HPD variant caused by the adjusted AM parameters could have been responsible for the global subgrain formation.

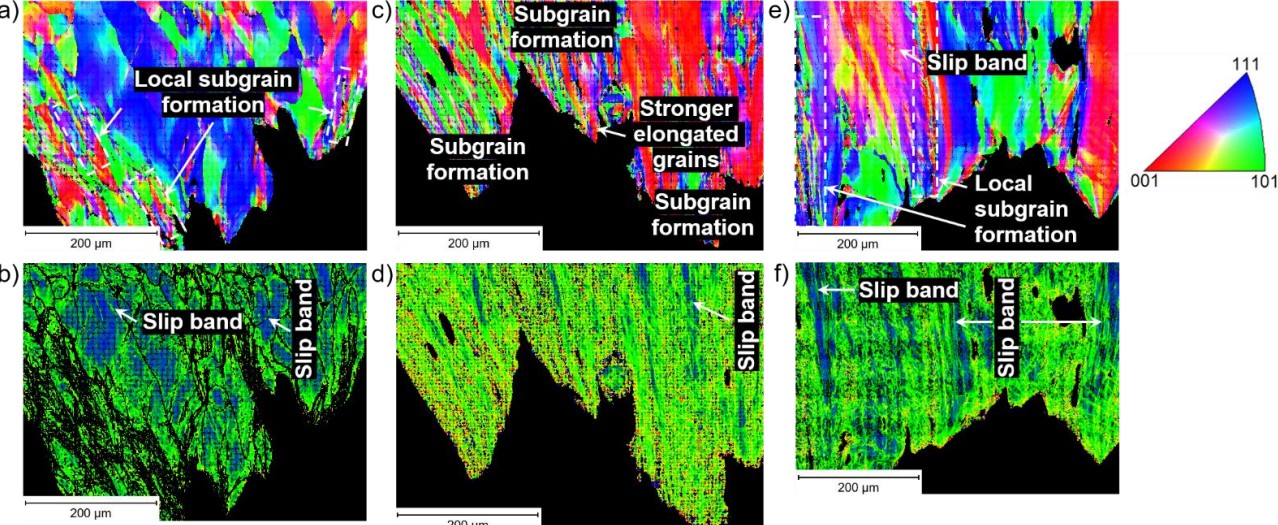

**Figure 13.** EBSD- and misorientation mappings taken longitudinally to the fracture surfaces after the tensile tests at 650 °C on HiperFer[AM] manufactured with (**a**,**b**) LPD and (**c**,**d**) HPD, as well as on (**e**,**f**) conventional HiperFer 17Cr2.

Effective grain refinement caused by the high degree of global sub-grain formation obviously yielded higher plastic deformability (Figure 13d) in the case of the HPD variant. From a comparison of Figure 13a,b (LPD), limited dislocation activity and/or plastic deformation in sub-grain free areas was evident, which indicates that deformation preferentially occurred due to sub-grain formation. This explained the significantly increased ductility of the specimens that were manufactured when applying HPD. In conventionally produced HiperFer 17Cr2 sheet material, local sub-grain formation occurred similarly to the LPD HiperFer[AM] (Figure 13a,e) variant, which explained the matching fracture elongations. Moreover, a stronger tendency toward slip band formation was observed in the conventionally produced HiperFer 17Cr2 (cf. Figure 13b,d–f).

Minimum creep rates of the HiperFer[AM] variants were determined with respect to the build direction (longitudinal (L) and transversal (T)) at 650 °C (Figure 14). In the compression creep experiments, the LPD-manufactured HiperFer[AM] specimens already exhibited unexpectedly high creep rates $> 10^{-4}\,\mathrm{h}^{-1}$ at a stress of 70 MPa, while much higher stress (130 MPa) was needed for the HPD specimens. The behavior that occurred in the LPD variant is referred to as "Power Law Breakdown (PLB)" [57]. At high stresses, higher strain rates occur more often than predicted by the power law relationship. This behavior is known as PLB. The cause for the PLB of LPD-manufactured HiperFer[AM] at comparatively low stress was a pronounced change in the grain structure: In the as-built state, the microstructure consisted of directionally solidified (Figure 15a) grains, which then changed to a globularly recrystallized microstructure (Figure 15b) during the execution of the

compression creep experiment. A directionally solidified microstructure obviously cannot effectively be stabilized by Laves phase precipitation and should therefore be avoided. In contrast, the microstructure of the HPD variant remained stable. Only the grain width of the directional grains between the globular ones tended to decrease (cf. Figure 15c,d). The increased creep strength of the HPD manufactured HiperFer$^{AM}$ was based on this stable microstructure. The building direction only had a minor impact on the minimum creep rates measured. Compared with HiperFer 17Cr2, the minimum creep rate in the first HPD-manufactured HiperFer$^{AM}$ variant (mechanically alloyed) was only about 0.5 to 1 order of magnitude higher (with a tendency of decreasing discrepancy toward higher stress (Figure 14)). This indicates a good potential for high-temperature application of additively processed HiperFer steel. The white dots in Figure 15a,c were incompletely melted tungsten clusters (cf. Figure 7)

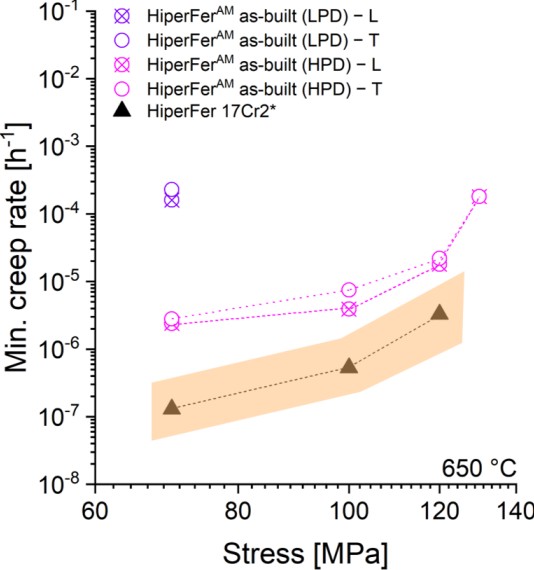

**Figure 14.** A 650 °C Norton plot of HiperFer$^{AM}$ with respect to the area-specific laser power and build direction in comparison to HiperFer 17Cr2* Data from [13].

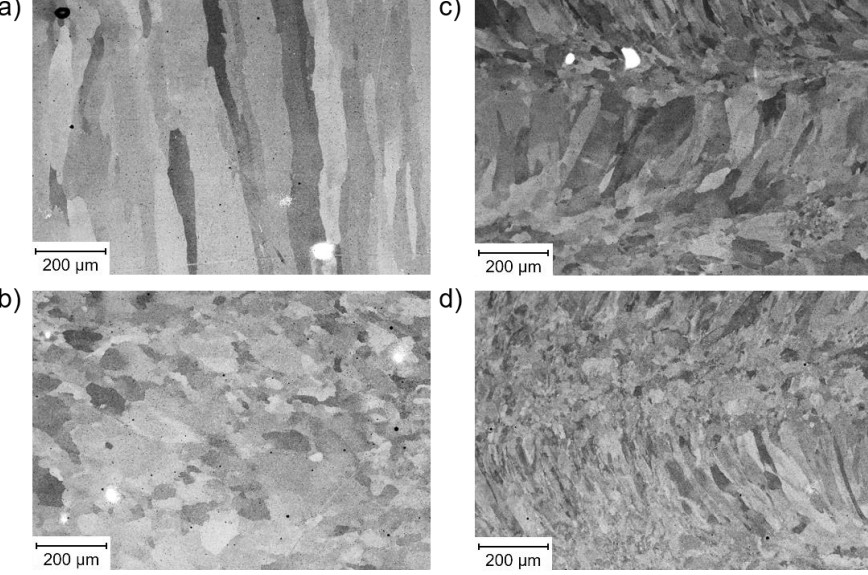

**Figure 15.** Comparison of as-built and post-compression creep experiment microstructures (specimens taken longitudinally to build direction). LPD: (**a**) as-built and (**b**) after compression creep. HPD: (**c**) as-built and (**d**) after compression creep.

Furthermore, the hardness increased after the compression creep testing, independently of the applied area-specific laser power and the building direction (Table 4). This indicates the high strengthening potential of HiperFer$^{AM}$.

## 4. Conclusions

The first high-chromium, Laves-phase-strengthened HiperFer$^{AM}$ alloy that benefits from the additive manufacturing process and counteracting process-related defects, such as pores due to thermomechanically triggered [9,33] Laves phase precipitation, was developed.

Specific adjustment of the AM process parameters directly allowed for controlling the microstructural parameters, such as the grain size/morphology, as well as the size and number of the Laves phase particles, and thus, the resulting mechanical properties. At ambient temperature, UTS values of ~667/~760 MPa (LPD/HPD) were achieved in the as-built condition, with an elongation at fracture of ~1/1.6%. Subsequent stress relief due to the annealing of HiperFer$^{AM}$ HPD resulted in a nearly tripled elongation at rupture and a UTS beyond 800 MPa. At 650 °C, the UTS of the HiperFer$^{AM}$ HPD variant was 55 MPa higher in comparison to the HiperFer 17Cr2 sheet material, with almost doubled elongation at rupture (~33.5%). Global sub-grain formation during plastic deformation was argued to be the reason for the increased elongation at rupture. In general, additive manufacturing was found to produce smaller grain sizes than conventional manufacturing ($\leq 200$/~500 μm).

Furthermore, the minimum creep rates of the HiperFer$^{AM}$ HPD variant ranged from approximately 0.5 to 1 order of magnitude higher than that of conventionally produced HiperFer 17Cr2. In contrast, the LPD variant exhibited premature power-law breakdown under creep loading, which was caused by instability of the directionally solidified as-built grain structure. Creep properties did not show a dependency on build direction.

This first mechanically alloyed HiperFer$^{AM}$ custom-made variant for the additive manufacturing process demonstrated high potential for high-temperature application. Furthermore, the substitution of comparatively expensive austenitic 316L steel in a small (special) series of ambient temperature components could be a realistic option. After grain size optimization and stress relief annealing, the resulting HiperFer$^{AM}$ material may offer a UTS of about 1 GPa at a reasonably lower cost.

## 5. Outlook

The presented HiperFer$^{AM}$ alloy will serve as a basis for further alloy development and optimization with a focus on tailored precipitation kinetics. To achieve this, thermokinetic simulations will be performed. In addition, the production of pre-alloyed powders for the LPBF process will be carried out. Further research will focus on the production of tailored microstructures with increased control over the in-build precipitation heat treatment and residual stress reduction by varying the AM process parameters and integrating substrate heating.

**Author Contributions:** Conceptualization, T.F. and B.K.; methodology, T.F., B.K. and X.F.; software, T.F., validation, T.F., B.K. and M.B.W., formal analysis, T.F., B.K., X.F. and M.B.W.; LMD manufacturing, M.B.W., investigation, T.F., B.K. and M.B.W., resources, B.K.; data curation, T.F., writing—original draft preparation, T.F.; writing—review and editing: B.K. and M.B.W.; visualization, T.F.; project administration, T.F. and B.K. All authors have read and agreed to the published version of the manuscript.

**Funding:** This research was funded by the German Helmholtz Society framework programme "Energy Efficiency, Materials and Resources".

**Institutional Review Board Statement:** Not applicable.

**Informed Consent Statement:** Not applicable.

**Data Availability Statement:** Not applicable.

**Acknowledgments:** The authors would like to acknowledge the support of B. Werner and H. Reiners for the mechanical testing, V. Gutzeit and J. Bartsch for the sample preparation, and E. Wessel and D. Grüner for performing the microstructural investigations.

**Conflicts of Interest:** The authors declare no conflict of interest.

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
