# Peer review of "Additive Manufacturing Potentials of High Performance Ferritic (HiperFer) Steels"

_applsci, doi:10.3390/app12147234_

Round 1
Reviewer 1 Report
Dear Authors,
Thank you for very interesting paper to review. The considered theme is very impressive and it could have impact on development of additive manufacturing processes of steels.
After reading of this manuscript, I have some questions and suggestions, which could improve the quality of paper.
1. I think that the investigations results of additive manufactured of steels should be extesively described.
2. Figure 2. For the presented fugures, the quantitative analysis of intragranullar Laves phaseprecipitates should be added. Because, the same binarized micrographs does not give the complete data about intragranular phase evolution.
3. What is the PFZ? It is not explained in the text. Please, add it.
4. If it is possible, you can add the strength curves (stress-elongation) for different considered alloys.
5. Figure 4 and 5. There is a typo (in the legend - "calculated"). Please, correct it.
6. The quality of figure 8 is very low. Please, improve it.
7. Figure 9. What are the black dots? Please, explain it. Moreover, the directional and globular grains should be marked with an arrow. It should be corrected.
8. There is no description of crystallographic orientation in Figure 10, 14 and 15 (EBDS images). Please, improve it.
9. Why the micrographs images in figure 11 were performed in the SE mode? I think that if you want to analyze the phase particles, the BSE mode of microstructure observations would be better.
10. What is the "At%" in the table 3? Is it elongation of sample?
11. What are the white dots in the figures 17 a,b,c? Is it the result of sample preparation process (grindind and polishing) or maybe another reason? It is not described in the text. Please, explain it.
12. The results of hardness can be rounded to the nearest whole number (table 4).
13. Some sentences are not understandable. The manuscript should be corrected and checked by native speaker.
Thank you for responses for questions in advance.
Best regards,
Reviewer
Reviewer 2 Report
Dear Authors,
The work is interesting. I did not find any significant errors or shortcomings. However, some corrections should be applied. In some results, the standard deviation is missing (I assume that a series of measurements was taken), e.g. Fig 12 (Area specific total number). It makes sense to add the standard deviation range in the chart. Then, for hardness measurements (Tab. 4). Another remark concerns figures 9 and 17. The images are not sharp. It is worth improving their quality.
Reviewer 3 Report
The paper “1816847” related to DED in AM was reviewed. Please follow the comments carefully and resubmit your paper for the next consideration and reviewing process.
1. Remove figures 1 and 2 from the introduction or if you want to keep them please justify them.
2. Figure 2 A is not clear. Please replace it.
3. How the experiment was designed. Add more detail.
4. The word of “Table 1” on page 4 shouldn’t be there.
5. Please check the typos.
6. Check page 6. It is blank.
7. How authors selected the process parameters for the experimentation?
8. The introduction needs to be updated by comparing the DED and Laser-based powder bed fusion LB-PBF which is also called SLM. Read and add the following new references.
· Evolution of temperature and residual stress behaviour in selective laser melting of 316L stainless steel across a cooling channel
· Fatigue life optimization for 17-4Ph steel produced by selective laser melting
· Proposal of design rules for improving the accuracy of selective laser melting (SLM) manufacturing using benchmarks parts
· Study of anisotropy through microscopy, internal friction and electrical resistivity measurements of Ti-6Al-4V samples fabricated by selective laser melting
9. Laser absorptivity in AM is important which shows the quality of the parts and transition from keyhole to conduction mode. Please read and add the following ref in this area. “The effect of absorption ratio on meltpool features in laser-based powder bed fusion of IN718”.
Round 2
Reviewer 1 Report
Dear Authors,
Thank you for all answers and corrections of manuscript. I accept this paper to publish in the presented form.
Best regards,
Reviewer
Reviewer 3 Report
The paper is ready to publish.